# Epigenetic Modifications in Alternative Splicing of LDLR pre-mRNA on Hypercholesterolemia Following Aerobic Exercise Training

**DOI:** 10.3390/ijms26094262

**Published:** 2025-04-30

**Authors:** Jinfeng Zhao, Peirun Yan, Yana Pang, Yuankun Dong, Xiangrong Shi

**Affiliations:** 1Institute of Physical Education, Shanxi University, Taiyuan 030006, China; 2UNT Health Science Center, Fort Worth, TX 76107, USA; 3Department of Neurobiology, Shanxi Medical University, Taiyuan 030001, China

**Keywords:** alternative splicing, cholesterol regulation, histone modifications, low-density lipoprotein receptor, recruitment model

## Abstract

This study investigated whether exercise training improved cholesterol metabolism through modifying alternative splicing of the low-density lipoprotein receptor (LDLR). Blood lipids and expressions of LDLR splice variants were compared between exercise-trained and non-trained young adults with normal and high cholesterol. The expression of LDLR splice isoforms were examined using RT-PCR and the histone H3K36me3 by CHIP-assay in mouse liver following a 13-week normal or high-cholesterol-diet combined with or without 8 weeks of aerobic exercise-training. The influence of histone modifications on LDLR alternative splicing was examined in HepG2 cells (human liver cell-line). Expression levels of LDLR deletions in exons 4 and 12 (LDLR-∆Exon4 and LDLR-∆Exon12) were significantly higher in the obese adults with high-cholesterol. These LDLR splice variants were significantly lower in the exercise-trained than non-trained group with normal cholesterol. Thirteen weeks of high-cholesterol feeding increased LDLR-∆Exon14 expression in mice, which was diminished after 8 weeks of exercise training. When H3-K36me3 or the MORF-related gene on chromosomes 15 were overexpressed and interfered, the levels of LDLR-∆Exon4 and LDLR-∆Exon12 expression in HepG2 cells were significantly augmented and inhibited, respectively. Hypercholesterolemia was associated with augmented expressions of LDLR splice variants in obese adults and following high-cholesterol diet in mice. Aerobic exercise training prevented and reversed the dyslipidemia-related alternative splicing of LDLR pre-mRNA. The histone modifications contributed to the alternative splicing.

## 1. Introduction

Cardiovascular disease (CVD) remains the primary cause of death worldwide. According to the World Health Organization, CVD was responsible for 17.9 million (32%) deaths in 2019. The death rate of CVD-related diseases is forecast to reach 23.6 million (40%) by 2030 [1,2]. Atherosclerosis is a primary risk factor for CVD and its development can be accelerated by hypercholesterolemia [3,4]. Therefore, the prevention and control of hypercholesterolemia are of great relevance in controlling the morbidity and mortality of dyslipidemia-related CVD.

Human low-density lipoprotein receptor (LDLR) includes five functionally distinct structural domains and the LDLR gene consists of 18 exons and 17 introns, as illustrated by the sketch diagram in Figure 1 [5,6]. The LDLR pathway is the most vital link in cholesterol metabolism by taking up cholesterol-rich low-density lipoprotein (LDL) in circulation, preventing its accumulation in the bloodstream and enabling the cholesterol entering the cells to promote cell proliferation and the synthesis of steroid hormones. About 80–90% of LDL is cleared in this process [7]. Alternative splicing is a post-transcriptional process in eukaryotes through which multiple distinct gene transcripts can be generated [8]. Current estimates based on deep sequencing approaches suggest that more than 95% of human genes undergo alternative splicing [9]. Studies by Tveten and his colleagues have shown that there are different variant spliceosomes of LDLR pre-mRNA and that two missing variant spliceosomes are strongly associated with hypercholesterolemia [10,11]. One is a deletion in exon 4, missing a total of 127 amino acid residues, and this region lies in the ligand-binding structural domain (encoded by Exon 2–6), where LDLR-∆Exon4 affects the ligand capacity. The other splice variant is LDLR-∆Exon12, with a deletion of 47 amino acid residues, and the deletion region resides in the epidermal growth factor (EGF) precursor structural domain (encoded by Exons 7–14), which belongs to the extracellular membrane structural protein that underpins LDLR. Increased LDLR-ΔExon12 is also associated with reduced cholesterol clearance [12,13].

Although tremendous progress has been made in pharmacologic research and development, there are limited clinically ideal approaches to preventing hypercholesterolemia. Abnormality in the LDLR gene can remarkably increase plasma cholesterol levels and cause hypercholesterolemia [14], since the LDLR pathway plays a vital role in cholesterol metabolism and LDL removal, which take up LDL into the liver and prevent their accumulation in the bloodstream [7]. Thus, upregulating LDLR gene expression and preserving LDLR would be effective measures to control LDL levels in plasma. LDLR degradation inhibitors, i.e., proprotein convertase subtilisin-kexin type 9 (PCSK9) inhibitors, block PCSK9 proteins from breaking down LDLR on the surface of hepatocytes. Consequently, more LDLRs carry LDL into the liver, which increases LDL clearance. However, the clinical use of PCSK9 inhibitors is inhibited by their cost [15,16]. Aerobic exercise seems to be emerging as an effective non-pharmaceutical preventive and therapeutic intervention to control hypercholesterolemia. Wilund et al. [17] found in gallstone mice that endurance exercise significantly increased hepatic LDLR expression. Puglisi et al. [18] confirmed in older adults that a 6-week walking exercise increased mononuclear LDLR transcription and activity, and decreased plasma LDL. Currently, however, the mechanism by which aerobic exercise lowers blood cholesterol through upregulating LDLR gene expression and preserving LDLR is incomplete. The present study aims to investigate the effects of aerobic exercise training with or without a high-cholesterol diet in mice and exercise trained vs. non-trained young adults with different body mass on the control of hypercholesterolemia via epigenetic modifications in the gene expression of LDLR and, consequently, phenotype. 

In this study, we compared blood plasma lipids and LDLR deletion levels of Exon4 and Exon12 in leukocytes from physically active or aerobic-exercise-trained people and less physically active or non-trained people with normal and high cholesterol levels, since these two missing variant spliceosomes are strongly associated with hypercholesterolemia [10,11]. Furthermore, we examined the effect of 8-week aerobic exercise training on the changes in plasma lipids and hepatic alternative splicing of LDLR pre-mRNA through histone modifications in mice fed with a 13-week normal diet or a high-cholesterol diet. We hypothesized that aerobic exercise training effectively improves lipid metabolism by regulating alternative splicing of LDLR pre-mRNA. Since the biological function of LDLR splice variants is altered compared to full-length (FL) LDLR, exploring the mechanism of LDLR alternative splicing helps provide a better understanding of its contribution to lipid metabolism. We added cholesterol to the extracellular fluid as an in vitro model of hypercholesterolemia and examined the mechanistic roles of local histone modifications, chromatin-binding proteins, and splicing factors in LDLR alternative splicing in HepG2 cells of the human liver cell lines.

## 2. Results

### 2.1. Expression of LDLR Splice Variants and LDL Cholesterol Concentration in Humans with Different Levels of Habitual Physical Activity

There was no group difference in gender proportion, age or height; however, body weight was significantly lower in the exercise-trained group and higher in the high-cholesterol group as compared to the normal group (Table 1). The levels of moderate to vigorous physical activity (MVPA) were more than doubled in the exercise training vs. non-trained normal group, and were significantly lower in the high-cholesterol obese group (Table 1). Furthermore, the concentrations of LDL and TC were significantly higher, whereas HDL was significantly lower in the high-cholesterol obese group, which was associated with increased expression levels of LDLR-∆Exon4 and LDLR-∆Exon12 in the leukocytes as compared to both the non-trained normal and exercise-trained groups (Table 1). Habitual physical activity increased HDL, which was significantly greater compared to both the non-trained normal group and the group with hyperlipidemia (Table 1). The expression levels of LDLR-∆Exon4 and LDLR-∆Exon12 were statistically lower in the exercise-trained than non-trained normal group, although the plasma concentrations of LDL and TC were not statistically different between two groups (Table 1). The expression levels of both LDLR-∆Exon4 and LDLR-∆Exon12 in these young subjects were significantly (*p* < 0.001) correlated with their LDL, TC, TG, and HDL. The correlation coefficients were 0.78, 0.93, 0.71, and −0.47 between LDLR-∆Exon4 and LDL, TC, TG, and HDL, respectively, and 0.59, 0.86, 0.75, and −0.62 between LDLR-∆Exon12 and LDL, TC, TG, and HDL, respectively. These data suggested that LDLR deletions or mutations of Exon4 and Exon12 could occur in healthy young subjects with normal BMI before the presence of dyslipidemia. Habitual physical activity seemed to have a positive impact by diminishing the alternative splicing of LDLR pre-mRNA in young adults. Figure 1 illustrates the human LDLR mRNA with full-length 18 exons along with the splicing isoform of LDLR-∆Exon4 and LDLR-∆Exon12, i.e., deletions of the 4th and 12th exons [19,20].

### 2.2. The Effects of High-Cholesterol Diet and Exercise Training on Weight and Lipid Levels in Mice

Forty male mice were randomly assigned to four groups (Figure 2) before the interventions. There was no statistical difference in baseline body weight or cholesterol levels among the four groups of the mice (Figure 3A). After 4 weeks of high cholesterol feeding, the body weights of groups 3 (H group—high-cholesterol diet) and 4 (HE group—high-cholesterol diet with exercise training) were significantly higher than those of group 1 (C group—normal diet). There was no significant difference between groups 3 and 4 with a 4-week high-cholesterol diet prior to exercise training. The weight did not change in Group 2 (CE group—normal diet with exercise training) as compared to Group 1. However, after 13 weeks of high cholesterol feeding, the weight was significantly greater in Group 3 than Group 1 (Figure 3A). Eight-week aerobic exercise training did not change the weight in Group 2 as compared to Group 1, but normalized the weight gained with high-cholesterol feeding evident by a significant lower body mass in Group 4 compared to Group 3 (H group). There was no difference in body mass between Group 4 and Group 1 after the 8-week exercise training interventions (Figure 3A).

High-cholesterol feeding for 13 weeks significantly increased LDL, TC, and TG, and decreased HDL in Group 3 (Figure 3B). This high-cholesterol diet-induced hypercholesterolemia was significantly reduced following 8-week aerobic exercise training in Group 4 (Figure 3B). There were no differences in blood plasma lipid levels between Group 1 (C) and Group 2 (CE) (Figure 3B). Both diet and exercise factors significantly affect LDL, TC and TG. However, HDL was only significantly affected by the diet factor.

### 2.3. The Effects of High-Cholesterol Diet and Exercise Training on Liver Tissue in Mice

The mouse liver on the normal diet (C) was ruddy, soft and elastic (Figure 3C), and there was no noticeable difference compared to that with normal diet combined with 8-week exercise training (CE). High-cholesterol feeding (H) increased lipid accumulation in the mouse liver, which was reversed by 8-week aerobic exercise training (HE). H&E staining indicated that the nuclei of hepatocytes were clearly visible, and the shapes of hepatocytes were regular and arranged in cords around the central vein in the mice with normal diet (C) or normal diet combined with aerobic exercise training (CE). The mouse hepatocytes after a 13-week high-cholesterol diet (H) were markedly swollen and loosely arranged, and apparently embedded with lipid droplets. These morphological changes associated with mouse hepatocyte swelling and lesions following a 13-week high-cholesterol diet seem to be remarkably mitigated after being combined with 8-week exercise training (HE) based on the H&E staining observation (Figure 3D).

### 2.4. The Effects of High-Cholesterol Feeding and Exercise Training on Alternative Splicing of LDLR Pre-mRNA in Mice

In mouse hepatocytes, the spliced variant skipped exon14 (Figure 3E–G). LDLR-ΔExon14 in mice resulted in a partial deficit of the structural domain of the epidermal growth factor (EGF), a part of the extracellular structural proteins supporting LDLR. Therefore, LDLR-ΔExon14 was likely a mechanism that contributed to the diminished function of hepatic LDLRs and cholesterol metabolism in mice.

There were no significant differences in the epigenetic modifications of LDLR between Group 1 (C group, control group with normal diet) and Group 2 (CE group, normal diet with exercise training) based on LDLR-ΔExon14 by RT-Real Time PCR from hepatocytes in mice (Figure 3G). These data suggest that 8-week aerobic exercise training did not significantly modify the structure and function of LDLR proteins in the mice with a normal diet. High-cholesterol feeding significantly augmented histone H3-K36me3 modifications of LDLR around the LDLR exon14 in Group 3 (H group). However, a high-cholesterol diet combined with 8-week aerobic exercise training remarkably attenuated the hypercholesterolemia-related elevation of LDLR-ΔExon14 expression in Group 4 (HE). Nonetheless, it remained higher as compared to the group with the normal diet combined without (C group) or with exercise training (CE group); see Figure 3G.

### 2.5. The Effect of Cholesterol on the Alternative Splicing of LDLR Pre-mRNA in HepG2

To determine whether the alternative splicing of LDLR pre-mRNA is mediated by cholesterol, both the full-length transcript and the alternatively spliced forms, LDLR-∆Exon4 and ∆Exon12, were examined in HepG2 cells of the human liver cell line after sterol depletion or the addition of cholesterol. HepG2 cells were incubated with different concentrations of lipoprotein deficient serum (LPDS) from normal (N), i.e., without LPDS, to 20% of LPDS in 5% as each level of LPDS (Figure 4A). Cholesterol depletion induced downregulations of the LDLR-∆Exon4 and ∆Exon12 transcripts, and both alterations reached a nadir at 15% LPDS (Figure 4A). In contrast, provisions of different concentrations of LDL-cholesterol (LDL-C) to HepG2 cells increased the LDLR-∆Exon4 and ∆Exon12 transcripts, both of which peaked with 75 μg/mL LDL-C (Figure 4B).

### 2.6. The Effect of Hypercholesterolemia on Histone Modification in HepG2

Based on the in vitro high-cholesterol cell model (Figure 4B), HepG2 cells were cultured by adding 75 μg/mL LDL-C to explore the association of high cholesterol with histone modifications and the potential causal role of histone modifications in LDLR pre-mRNA alternative splicing. CHIP analysis was used to map a set of histone modifications across the alternatively spliced region in LDLR and to compare the differences between adding LDL-C 75 μg/mL (H) and the normal (N), i.e., control condition (normal = 0 LDL-C). Enriched plasma cholesterol with an addition of 75 μg/mL LDL-C remarkably changed the histone modifications around exon 4 and exon 12 of LDLR (Figure 4C,D). The levels of H3-K36me3 were distinctly enriched around the LDLR exon 4 and 12 as compared to the control or normal group. These results seemed to agree with the mouse data; that H3-K36me3 with the alternative exon around the region was augmented in association with hyperlipidemia following a 13-week high-cholesterol diet (Figure 3).

### 2.7. Transfection on HepG2 Cell: H3-K36me3 Regulated the Alternative Splicing of LDLR Pre-mRNA

Since H3-K36me3 seemed to be most prominently enriched around alternatively spliced exons on LDLR in both mouse hepatocytes and HepG2 cells, we transfected the H3-K36 methyltransferase plasmid into HepG2 cells to overexpress the transcript (overexp-H3K36) and measured the LDLR-∆Exon4 and ∆Exon12 transcripts with RT-PCR to investigate whether histone modifications were involved in alternate splice site selection (Figure 5). We found that LDLR-∆Exon4 and ∆Exon12 transcripts were significantly increased compared to the vector-transfected cells (Figure 5A–C). Conversely, down-regulation of the H3-K36 methyltransferase (sh-H3K36) by RNA interference (RNAi) significantly decreased the expression of LDLR-∆Exon4 and ∆Exon12 transcripts (Figure 5D–F). These data provided evidence that epigenetic modifications with overexpressed or interfered H3-K36me3 could augment or downregulate the alternative splicing of LDLR pre-mRNA of HepG2 cells.

### 2.8. Transfection on HepG2 Cell: MRG15 Regulated the Alternative Splicing of LDLR Pre-mRNA

Polypyrimidine tract-binding protein (PTB) is a splicing regulator, which binds to the LDLR exons [21] and promotes alternative splicing. PTB knock-down is associated with the repression of LDLR alternative splicing or the reduced expressions of LDLR-∆Exon4 and ∆Exon12, which suggests that the alternative splicing of LDLR seems to have PTB involved through binding to the silencing elements around exon 4 or 12 [11]. To define the molecular mechanism by which histone marks influence the splice site choice, we focused on the overexpression (overexp-MRG15, Figure 5G–I) and knockdown (sh-MRG15, Figure 5J–L) of the histone tail-binding protein MRG15, which was assumed to act as an epigenetic modulator through binding to H3-K36me3 and recruiting PTB [22] to affect the expressions of LDLR-∆Exon4 and ∆Exon12 transcripts, respectively. Our data suggested that both expressions of LDLR-∆Exon4 and ∆Exon12 transcripts were augmented by overexpressed MRG15 (Figure 5G–I) or downregulated by interfered MRG15 (Figure 5J–L), which confirmed MRG15 as a modulator of the alternative splicing LDLR pre-mRNA in HepG2 cells.

## 3. Discussion

The present study demonstrated that augmented variant spliceosomes of LDLR pre-mRNA were associated with hypercholesterolemia in young human adults. LDLR-Exon∆4 and LDLR-Exon∆12 mRNA expressions in human leukocytes were significantly higher in obese subjects with high cholesterol and lower in exercise-trained as compared to non-trained healthy young counterparts with normal cholesterol. Furthermore, hypercholesterolemia resulting from a 13-week high-cholesterol diet increased the methylation of H3K36 around the LDLR alternative exon 14 in the hepatocytes of mice. This augmented expression of hepatic LDLR-Exon∆14 associated with dietary-induced hypercholesterolemia in mice was significantly reversed following 8 weeks of aerobic exercise training. Collectively, these data suggest that hypercholesterolemia or dyslipidemia and augmented variant spliceosomes of LDLR pre-mRNA could be closely interrelated and likely reinforced to form a vicious cycle. However, alternative splicing of LDLR pre-mRNA associated with hypercholesteremia in obese human adults or high-cholesterol feeding in mice could be effectively prevented or reversed along with improved cholesterol metabolism by a physically active lifestyle or aerobic exercise training. Using a hypercholesterolemic model in HepG2 cells of the human liver cell line, we identified that the expression levels of H3K36me3 around LDLR exons 4 and 12 were significantly increased. These changes in the histone marker of H3K36me3 in HepG2 cells were paralleled by the changes in chromatin-binding protein MRG15 evident by transfection-induced overexpression and knockdown processes. We propose that the mechanism of alternative splicing of LDLR pre-mRNA seemed to act through an adapter system mediated by H3-K36me3-MRG15 as an integral part of the splicing regulator.

### 3.1. Physically Active Lifestyle Prevents the Overexpressed Alternative Splicing of LDLR Pre-mRNA

Low-density lipoprotein receptors (LDLR) bind to LDL particles in the blood mediated by the ligand Apolipoprotein B100, which is degraded by the enzyme in the liver. Then, LDLR returns to the hepatic cell membrane for reuse and binding to LDL particles, thereby reducing circulating cholesterol levels [23,24]. Mutations in the LDLR gene interrupt its activity and result in hypercholesterolemia [25,26]. It has been shown that the deletions of LDLR exons 2–6 and exons 11–12 are associated with human hypercholesterolemia [27,28]. The present study demonstrated that the expression levels of LDLR-∆Exon4 and LDLR-∆Exon12 were significantly elevated in the obese hypercholesterolemic young adults, which could explain the higher levels of blood plasma LDL (Table 1). Aerobic exercise is widely considered to be an effective way to lower blood cholesterol, which is associated with lowered LDLR degradation inhibitor PCSK9 and therefor promotes induced LDLR transcription and expression [29,30,31]. Our data indicated that the expression levels of LDLR-ΔExon4 and ΔExon12 were significantly lower in the exercise-trained than the non-trained subjects with normal or high cholesterol levels. It is worth noticing that the expression levels of LDLR-ΔExon4 and ΔExon12 were lower in healthy young adults with normal BMI and cholesterol level compared to the obese counterparts, but higher when compared to the physically active or exercise-trained counterparts (Table 1). These data suggested that overexpressed alternative splicing of LDLR pre-mRNA could occur in healthy young adults with normal BMI prior to the appearance of hypercholesteremia and/or obesity, indicating augmented levels of LDLR-ΔExon4 and ΔExon12 leading to impaired cholesterol metabolism. Nonetheless, this potential detrimental mutation in LDLR pre-mRNA in healthy young adults who had normal BMI and cholesterol levels could be prevented by habitual physical activity or a physically active lifestyle based on this preliminary cross-sectional study (Table 1).

### 3.2. Aerobic Exercise Training Reverses High-Cholesterol Diet-Induced Alternative Splicing

The increased alternative splicing of LDLR mRNA led to hypercholesteremia. Furthermore, the present study also demonstrated that high-cholesterol feeding-induced hypercholesteremia could augment the detrimental mutation in LDLR, evident by the increased expression level of hepatic LDLR-Exon∆14 in mice after a 13-week high-cholesterol diet (Figure 3). However, 8-week aerobic exercise training significantly reversed the high-cholesterol diet-induced hypercholesteremia and augmented the expression level of hepatic LDLR-Exon∆14 in mice. Interestingly, there was no significant difference in body weight, blood plasma lipid concentrations, or alternative splicing of LDLR mRNA between the exercise-trained and non-trained groups with normal diet in mice. This suggested that a prolonged duration of more than 8 weeks of exercise training may be needed to produce a more complete and profound impact, although 8-week exercise training was significantly sufficient, although not entirely, to reverse the 13-week high-cholesterol diet-impaired function of cholesterol metabolism, weight gain, and increased expression level of hepatic LDLR-Exon∆14 in mice. This longitudinal comparison in cohort mice demonstrated that aerobic exercise training was protective against high-cholesterol diet-induced overexpression of hepatic LDLR alternative splicing and hypercholesterolemia. In addition, data from the present study suggested that dietary selection was also a critical factor influencing body mass, blood plasma lipids and LDLR alternative splicing.

### 3.3. The Effect of Histone Modification on Alternative Splicing of LDLR Pre-mRNA

Many histone modifications are present at high densities in exonic nucleosomes. Interactions between RNA-binding proteins (RBPs) and histone modifications, particularly with elevated H3K36me3 in these nucleosomes, can increase the local concentration of RBPs in exonic splice sites and lead to alternative splicing [32]. There is growing evidence that cholesterol homeostasis is regulated by epigenetic mechanisms such as histone acetylation and DNA methylation, which exerts an essential function in the synthesis, elimination, transport, and storage of cholesterol [33]. Human LDLR-ΔExon4 and LDLR-ΔExon12 are closely associated with hypercholesterolemia. Drug interventions can regulate the alternative splicing of LDLR pre-mRNA and increase normal LDLR transcripts in hypercholesterolemic patients by affecting the expression of histone methylation and acetylation and controlling hypercholesterolemia [34,35,36]. It has recently been found that aerobic exercise increased human telomerase reverse transcriptase (hTERT) gene expression and altered full-length hTERT splicing in contractile tissues, and may help maintain the telomere length necessary to improve the function and health of the organism [37]. To explore how the underlining mechanism of LDLR pre-mRNA alternative splicing on cholesterol metabolism or homeostasis was affected by aerobic exercise training, we examined LDLR histone modifications in mice. We found that hepatocytic H3-K36me3 was elevated around LDLR-Exon14 in hypercholesterolemic mice, and this change was markedly reversed after aerobic exercise training. We further overexpressed and interfered with H3K36me3 methyltransferase in HepG2 cells and found that LDLR-ΔExon4 and LDLR-ΔExon12 expression were significantly increased after H3K36me3 methyltransferase overexpression and robustly decreased after H3K36me3 methyltransferase interference. These data suggest that H3-K36me3 influences the alternative splicing of LDLR pre-mRNA. The inhibition of H3K36me3 overexpression can prevent the alternative splicing of LDLR pre-mRNA and improve or reverse hypercholesteremia level.

### 3.4. The Effect of Genomic Recruitment on Alternative Splicing of LDLR Pre-mRNA

MORF-related gene on chromosomes 15 (MRG15) is a transcription factor containing the chromosomal structural domain of a methyl lysine reader with two conserved structural domains, the N-terminal chromosomal domain and the C-terminal MRG structural domain. The chromosomal domain recognizes H3K36me3 signals within active transcribed genes and recruits multiple chromatin-associated complexes that regulate gene transcription, DNA repair and RNA splicing [38,39]. It has been shown that MRG15 coordinates epigenetic remodeling to regulate hepatic lipid metabolism [40]. We postulated that increased LDLR H3-K36me3 could be recognized by MRG15, which in turn could recruit polypyrimidine tract-binding protein (PTB) to affect the alternative splicing of LDLR pre-mRNA in hypercholesterolemia (Figure 6). Our experimental protocols overexpressed and interfered with the methyltransferase of H3K36me3, demonstrating that H3K36me3 could affect the alternative splicing of LDLR pre-mRNA. To further test our conjecture, we next overexpressed and interfered with MRG15 in HepG2 cells and found that the expressions of LDLR-ΔExon4 and LDLR-ΔExon12 showed significant upregulation and downregulation with MRG15 overexpression and interference, respectively. These data demonstrated that H3K36me3 around exons 4 and 12 of the LDLR alternative splice site could be recognized by MRG15, which could affect the selective splicing of LDLR pre-mRNA (Figure 6). MRG15 is one of the key factors regulating the alternative splicing of LDLR pre-mRNA through which cholesterol metabolism is affected.

### 3.5. Study Limitations and Perspectives

In the present study, there were no direct data on polypyrimidine tract binding protein (PTB), a member of the heterogeneous ribonucleoprotein family, correlated with changes in H3-K36me3 and MRG15, although it has been postulated to be involved in the alternative splicing of LDLR pre-mRNA [11,21] and almost all steps of mRNA metabolism that act rapidly between the nucleus and cytoplasm [41]. PTB negatively regulates the selective splicing of fibroblast growth factor receptor 2 (FGFR2), which contains two mutually exclusive exons (IIIb and IIIc), with two splice variants, FGFR2-IIIb (expressed in epithelial cells) and FGFR2-IIIc (expressed in mesenchymal cells). Analysis of FGFR2 histone modifications showed that H3-K36me3 is enriched on FGFR2 gene mesenchymal cells, which is recognized by the MRG15 protein that recruits PTB to the silencing element around exon (IIIb) of FGFR2 gene mesenchymal cells, and leads to the excision of exon IIIb and thus affects the selective splicing of pre-mRNA [22,42]. Likewise, polypyrimidine tract binding protein 1 (PTBP1) induces the alternative splicing of genes involved in the maintenance of cholesterol homeostasis as LDLR and 3-hydroxy-3-methylglutaryl-CoA reductase, which are relevant in the modulation of cellular cholesterol biosynthesis and uptake. It has been demonstrated that PTB knockdown reduces the expression levels of LDLR-ΔExon4 and ΔLDLR-ΔExon12 [11], indicating that LDLR alternative splicing binds to the silencing element around exon 4 or 12 via PTB protein, leading to an inhibited expression of LDLR [11,21]. We proposed an adaptor system of H3-K36me3-MRG15-PTB involved in the alternative splicing of LDLR pre-mRNA, during which increased levels of H3-K36me3 around LDLR exons 4 and 12 were recognized by MRG15 which, in turn, recruited PTB protein to affect the alternative splicing of LDLR pre-mRNA in hypercholesterolemia (Figure 6).

The present study only examined the morphological changes in the mouse livers with normal or high cholesterol diet after 8 weeks of aerobic exercise. Future studies are needed to examine the effect of aerobic exercise training on the steatosis associated with hyperchloremia and the expression levels of H3-K36me3-MRG15-PTB along with splicing variants of LDLR pre-mRNA in normal human hepatocytes. In addition, longitudinal studies on humans stratified by age, gender and ethnicity, and more evidence on the time duration of the protective exercise training effect on body mass, blood plasma lipids and LDLR function with de-training, are needed in the field.

Eight-week aerobic exercise training significantly reversed the 13-week high-cholesterol diet-induced body mass gain, hypercholesterolemia, and splicing variants of LDLR pre-mRNA in mouse hepatocytes. Prolonged training duration seems needed to elicit a more complete profound impact and to determine the dose–response effect on high-cholesterol diet-induced changes in body weight, hypercholesterolemia and splicing variants of LDLR pre-mRNA. Nonetheless, our data suggest that aerobic exercise training can effectively prevent or interrupt the vicious circle between augmented splicing variants of LDLR pre-mRNA and hypercholesterolemia induced by a physically inactive lifestyle or unhealthy dietary choices.

In conclusion, the present study demonstrates that overexpressed levels of LDLR alternative splicing are significantly associated with increased blood plasma cholesterol levels and obesity. This augmented LDLR alternative splicing may occur prior to the appearance of hypercholesterolemia and/or obesity in healthy young adults. On the other hand, high-cholesterol diet-induced hypercholesterolemia may augment the expression levels of LDLR alternative splicing. Aerobic exercise training significantly prevents hypercholesterolemia by modulating alternative splicing of LDLR pre-mRNA. The mechanism of hypercholesterolemia induced overexpressed histone modifications in LDLR alternative splicing, which is likely mediated through an adapter system which involves H3-K36me3 and MRG15.

## 4. Materials and Methods

### 4.1. Human Subjects

All participants gave written informed consent and passed a physical screening that included a questionnaire on medical/health and physical activity history. These healthy young subjects had no cardiovascular, respiratory or renal disease. The study protocol complied with local and international ethical guidelines on the use of human subjects and was approved by the ethical committee at the Shanxi University (Protocol #SXULL2020075, approved on 25 December 2020). Blood samples from young adults with normal (7 men and 3 women) and high (8 men and 2 women) cholesterol levels were obtained from the blood bank of the Taiyuan Liankang Physical Examination Center. In addition, venous blood samples from 10 college students (7 men and 3 women) who had engaged in long-term aerobic exercise, i.e., cycling and/or running (≥2 years), from the College of Physical Education at the Shanxi University were collected as the exercise-trained group. The levels of moderate to vigorous physical activity (MVPA) were assessed based on the answers to the questions “On average, how many days a week did you engage in physical activities such as jogging, cycling, climbing stairs, etc.” and “On average, how many minutes did you engage in physical activities at this level”. Table 1 summarizes the general information of these participants.

### 4.2. Assessment of Blood Plasma Lipid Levels

Blood samples of the human subjects (and mice) were centrifuged at 12,000 rpm for 1 min at 4 °C. Clear portions of the blood plasma were extracted and stored at −20 °C until analyzed. Plasma total cholesterol (TC), triglyceride (TG), low-density lipoprotein (LDL), and high-density lipoprotein (HDL) were measured by a microplate reader using an assay kit (Jianchen Bioengineering Institute, Nanjing, China).

### 4.3. Human Blood RNA Extraction

Human blood samples (in a purple tube) were mixed with erythrocyte lysate and left at room temperature for 10 min. Then, the treated blood samples were centrifuged at 10,000 rpm for 1 min and the precipitation of leukocytes was collected. The collected leukocytes were stored at −20 °C until analyzed for human LDLR proteins and mRNA.

### 4.4. Animals and Exercise Training Protocol

Forty male 8-week-old C57BL/6J mice were purchased from the China Institute for Radiation Protection, and randomly assigned to 4 groups: Group 1 (n = 10)—control (C) group with normal diet; Group 2 (n = 10)—control with exercise (CE) group, i.e., exercise training group with normal diet; Group 3 (n = 10)—high-cholesterol diet (H) group; and Group 4 (n = 10)—high-cholesterol diet combined with exercise (HE) group (Figure 2). The animals were housed in groups of 5 in standard polycarbonate cages and the vivarium conditions were maintained at 22 °C (humidity ~60%) under a 24 h light (06:00–18:00)–dark (18:00–06:00) cycle. The high-cholesterol diet purchased from Beijing Botai Hongda Biotechnology Co., Ltd. (Beijing, China) consisted of 24.2% protein, 40.1% carbohydrate, 27.4% fat, 5.8% fiber and 2.5% minerals (48.5% kcal from fat). The normal diet was provided by the Chinese Experimental Animal Resources Research Institute for food and drug control (10% kcal from fat). The treadmill exercise was set to a running platform incline of 6%, speed of 26 m/min, 45 min/session, 1 session/day, and 6 sessions/week for 8 weeks. In addition, there was a phasing-in week before starting the dietary regimen and exercise training, respectively (Figure 2). After twenty-four-hour fasting from the last day of the protocols, the mice were weighed and euthanized. Blood was collected and the liver was harvested. The experimental protocol was reviewed and approved by the Institutional Animal Care and Use Committee of the Shanxi University (Protocol #SYXK-JIN 2018-0005, approved on 25 December 2020), conforming to the local and international guidelines for the ethical use of animals. 

### 4.5. Hematoxylin–Eosin Staining (H&E Staining)

Part of the liver tissue was fixed in 10% formalin, dehydrated, and embedded in paraffin with the slicer-cut sections of liver tissue to a thickness of approximately 6 μm. After dewaxing and H&E staining for 5 min, the stained cell nuclei were rinsed with tap water and returned to blue with phosphate-buffered saline (PBS), followed by eosin staining for 2 min and rinsing. After staining, the tissues were immersed in gradient alcohol for 5 min to dehydrate and in xylene I and II for 5 min until transparent and were sealed with neutral glue. Morphological changes in the liver were examined under a microscope; the number of lipid droplets in the tissue sections was analyzed using Image J software (version 1.53e) and stained with hematoxylin and eosin (H&E) (Solarbio, Shanghai, China).

### 4.6. Cell Culture

HepG2 cells of the human liver cell line were obtained (National Infrastructure of Cell Line Resource) and grown in Dulbecco’s modified Eagle’s medium (DMEM) supplemented with 10% fetal bovine serum (FBS), streptomycin (100 g/mL) and penicillin (100 U/mL) at 37 °C with 5% CO_2_. The cells were seeded in 6- or 12-well plates and grown to 80–90% confluence before use. The cells were exposed in replicate to conditions of sterol depletion that were treated with lipoprotein-deficient serum (LPDS) at different concentrations (5%, 10%, 12%, 15% or 20%) and sterol accumulation with LDL-C (5, 10, 25, 50, 75 or 100 μg/mL) for 24 h. HepG2 cells have been extensively utilized as an in vitro model for studying cholesterol metabolism, such as through constructing a hyperlipidemic hepatocyte model to investigate the role of Anti-b in regulating abnormal TC and TG accumulation in vitro [43] and to validate the mechanism of action of AKG in vitro that effectively attenuates lipid accumulation, ameliorates oxidative stress and regulates mitochondrial function in hepatocytes [44]. Recent studies have demonstrated that atorvastatin (ATS) increased the fold enrichment of H3K4me3 at the PCSK9 promoter in the HepG2 model by using HepG2 as a model [45] and the κ-keratinase-produced KCOs significantly decreased TC, TG and LDL levels and increased HDL levels in HepG2 cells [46]. These data seem to support the validity of using the classical cell model HepG2 for the relevant studies. In the present study, these HepG2 cells were applied to investigate the cholesterol-regulated alternative splicing of LDLR pre-mRNA and the derived effects on histone modification.

### 4.7. Plasmids and Transfection

Plasmids of overexpress-H3K36me3 and sh-H3K36me3 and overexpress-MRG15 and sh-MRG15 were purchased (GeneCopoeia, Guangzhou, China). HepG2 cells were seeded at 2.5 × 10^4^ cells in 500 μL of an appropriate culture medium containing serum and antibiotics. On the day of transfection, dilute 0.4 μg DNA was dissolved in Tris-EDTA (TE) buffer, pH 7–8, with medium and without serum, proteins or antibiotics to a total volume of 60 μL. All the plasmids were transfected with Attractene Transfection Reagent (Qiagen, Shanghai, China). Transfections were performed according to the standard manufacturer’s protocol at a 3:1 Attranctene to DNA ratio.

### 4.8. RNA Analysis

The tissues extracted from human leucocytes and mouse liver samples or HepG2 cells were assessed for total RNA using Trizol kits (Invitrogen, Thermo Fisher Scientific, Shanghai, China) following the manufacturer’s protocol. Total RNA was reverse-transcribed into cDNA using PrimeScript RT Master Mix (TaKaRa, Dalian, China). Quantitative PCR was performed using an SYBR Premix Ex TaqTMIIon and Light Cycler96 quantitative fluorescence PCR instrument according to the standard protocols. The thermal protocols consisted of (1) denaturing for 30 s at 95 °C, (2) 40 cycles of 95 °C for 5 s, and (3) annealing at 60 °C for 34 s. Table 2 summarizes the primers used for the analyses of mouse and human LDLR RNA in the study.

### 4.9. Chromatin Immunoprecipitation (CHIP)

CHIP analysis was carried out with a CHIP assay kit (Beyotime Institute, Haimen, China) according to the manufacturer’s protocols. Briefly, 1% formaldehyde was added directly to the cells, followed by incubation at 37 °C for 10 min to cross-link the target protein and the corresponding genomic DNA, and 1.1 mL Glycine Solution (10×) was added to stop the cross-linking reaction. Chromatin samples were sheared by sonication to break the DNA mostly to 200–1000 bp size and used to immunoprecipitate fragments of chromatin with histone antibody (Abcam, WuXiDiagnostics, Shanghai, China). Finally, DNA was recovered after reversed cross-links between protein and DNA, and CHIP-PCR results were obtained through PCR reaction and DNA agarose electrophoresis. The primer sequences are listed in Table 2.

### 4.10. Statistical Analysis

One-factor analysis of variance (ANOVA) was applied to determine the significance of the group factor in the physical data, plasma lipids and expression levels of LDLR-∆Excon4 and LDLR-∆Excon12 among three groups: non-trained groups with normal vs. high-cholesterol vs. exercise-trained group, or the lipid-treatment factor for the cholesterol-regulated intracellular data. Two-factor ANOVA was applied to determine the dietary factor (normal diet vs. 13-week high-cholesterol diet) and the exercise factor (with vs. without 8-week aerobic exercise training) on the mouse data. Post hoc analysis (Tukey’s test) was performed if the major factor(s) were significant at *p* < 0.05. Statistical analyses were performed using SPSS 26.0 software. Data are expressed as means ± standard error (SE) of the mean. A *p*-value of 0.05 was set as the threshold for statistical significance. All measurements of human blood and mouse tissues were determined in triplicate.

## Figures and Tables

**Figure 1 ijms-26-04262-f001:**
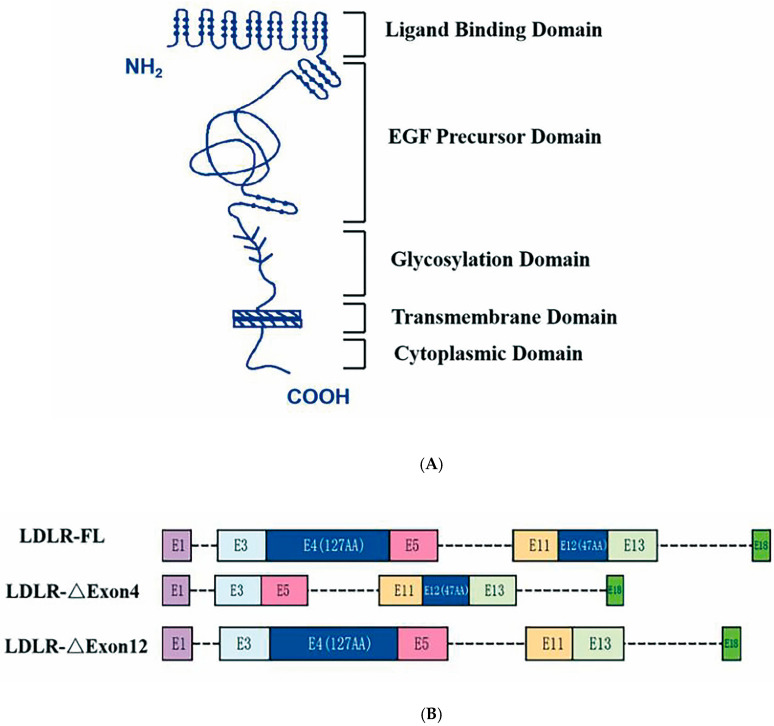
Sketch of low-density lipoprotein receptor structural domains and mRNA. Panel (**A**): Five structural protein domains of human low-density lipoprotein receptor (LDLR)—ligand binding structural domain (encoded by exons 2–6), epidermal growth factor (EGF) precursor domain (encoded by exons 7–14), glycosylated domain (encoded by exons 15), a transmembrane domain (encoded by exons 16 and 17), and a cytoplasmic domain (encoded by partial sequences of exons 17 and 18). Morphological changes in LDLR associated with ΔExon4 and ΔExon12 cause functional impairment of the ligand binding domain and EGF precursor domain, respectively. Panel (**B**): LDLR full length (FL) of human leukocyte with E1 to E18 exons and 17 introns, and the splicing isoform of LDLR-∆Exon4 and LDLR-∆Exon12, respectively.

**Figure 2 ijms-26-04262-f002:**
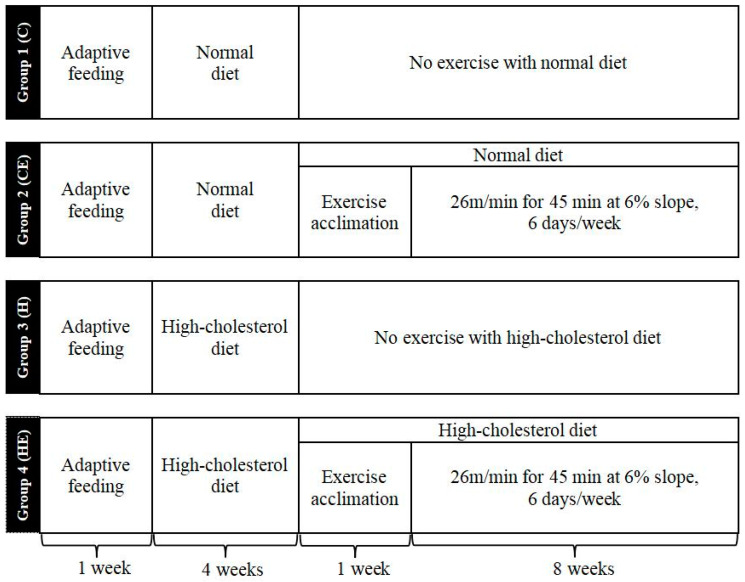
Overview of the experiment protocol for mice. Group 1 or C: control group (i.e., normal diet); Group 2 or CE: control with exercise training for 8 weeks; Group 3 or H: high-cholesterol diet for 13 weeks; Group 4 or HE: high-cholesterol diet combined with exercise training. Eight-week treadmill exercise training started after 4-week high-cholesterol diet and 1-week exercise phase-in with high-cholesterol diet for Group 4 (HE group). The animals were euthanized after 24 h fasting from the last day of the protocols.

**Figure 3 ijms-26-04262-f003:**
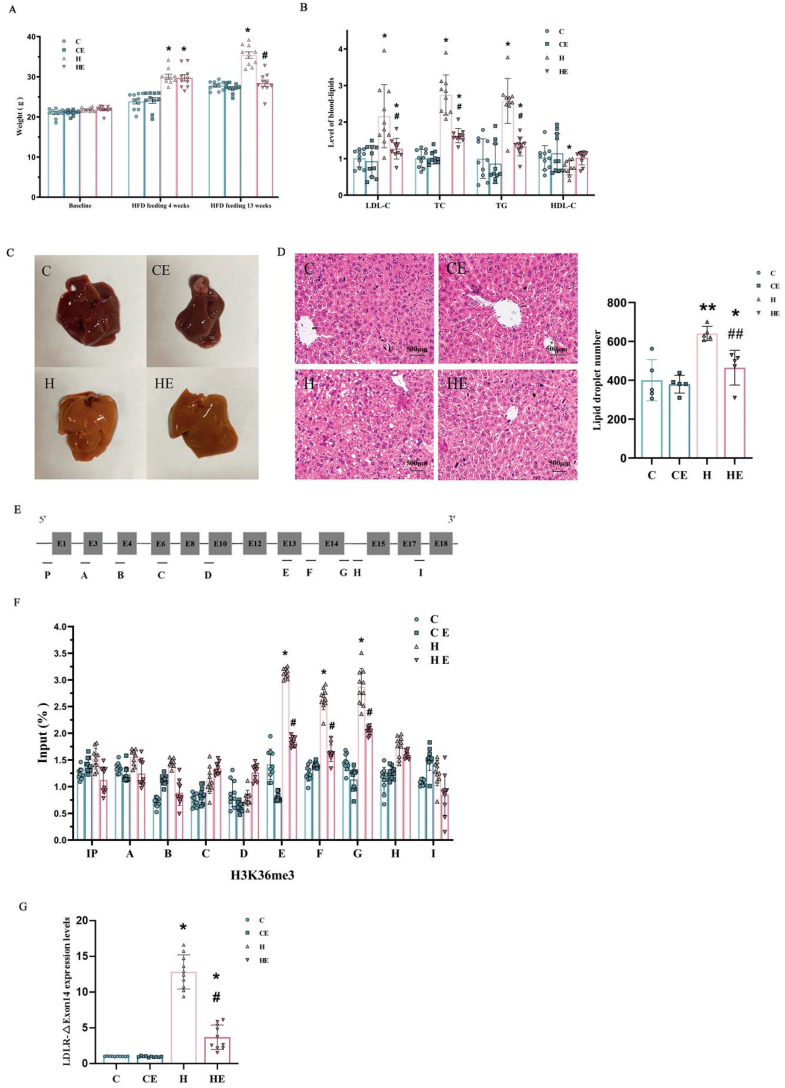
Effects of high cholesterol feeding and aerobic exercise training on weight, blood lipids, hepatic steatosis and histone modification in mice. Panel (**A**): Baseline body weight, weight after 4-week and 13-week high-cholesterol diet combined without or with 8-week exercise training. There was no difference in baseline weight among the groups. After a 4-week high-cholesterol diet, mice in Groups 3 (H group) and 4 (HE group) gained weight significantly as compared to Group 1 (C group). There was no significant difference in the weight between Group 2 (CE group) and Group 1 (C group). After a 13-week high-cholesterol diet, Group 3 (H group) further increased in weight. However, Group 4 (HE group) substantially reduced the weight gain, which was significantly less as compared to Group 3 (H group), and not different from the weight in Group 1 (C group). However, 8-week exercise training with normal diet had no significant effect on the weight in mice. Panel (**B**): Blood lipid level of C group was taken as 1 unit. Both diet and exercise factors significantly affected low-density lipoprotein-cholesterol (LDL-C), total cholesterol (TC) and triglyceride (TG). However, high-density lipoprotein-cholesterol (HDL-C) was significantly affected by the diet factor only, and not the exercise factor. Panel (**C**): Visual comparison of the liver tissues from the mice with control or normal diet (C), control diet after 8-week exercise training (CE), after 13-week high-cholesterol diet (H), and 13-week high-cholesterol diet combined with 8-week exercise training (HE), respectively. Panel (**D**): Hematoxylin–eosin (H&E) staining of liver sections from the different mice in each of these four groups, i.e., control or normal diet (C), normal diet with exercise (CE), high-cholesterol diet (H), and high-cholesterol diet with exercise training (HE). Panel (**E**): LDLR Full-Length mRNA of the mouse hepatocyte from RT (Real Time) PCR. Panel (**F**): Histone H3-K36me3 modifications of the hepatic LDLR. The effect of high-cholesterol diet appeared to be concentrated on the hepatic LDLR between the 13th and 14th exons of mice. Two-factor ANOVA indicated that both diet and exercise factors significantly affected the expression levels of LDLR-∆Exon14 mRNA. Panel (**G**): Relative expression levels of LDLR-∆Exon14 mRNA in the liver of mice in each group (the control group as 1). Individual data and group mean with SE bars, respectively, were reported for the control group with normal diet (C group), normal diet group combined with 8-week exercise training (CE group), high-cholesterol diet for 13 weeks (H group) and 13-week high-cholesterol diet combined with 8-week exercise training (HE group). * Indicates a significant difference when compared to the control (C) group and ^#^ indicates a significant difference when compared to the high-cholesterol diet (H) group, respectively. ** Indicates a significant difference when compared to the control (C) and normal diet with 8-week exercise training (CE) groups and ^##^ indicates a significant difference when compared to the normal diet with 8-week exercise training (CE) and high-cholesterol diet (H) groups, respectively.

**Figure 4 ijms-26-04262-f004:**
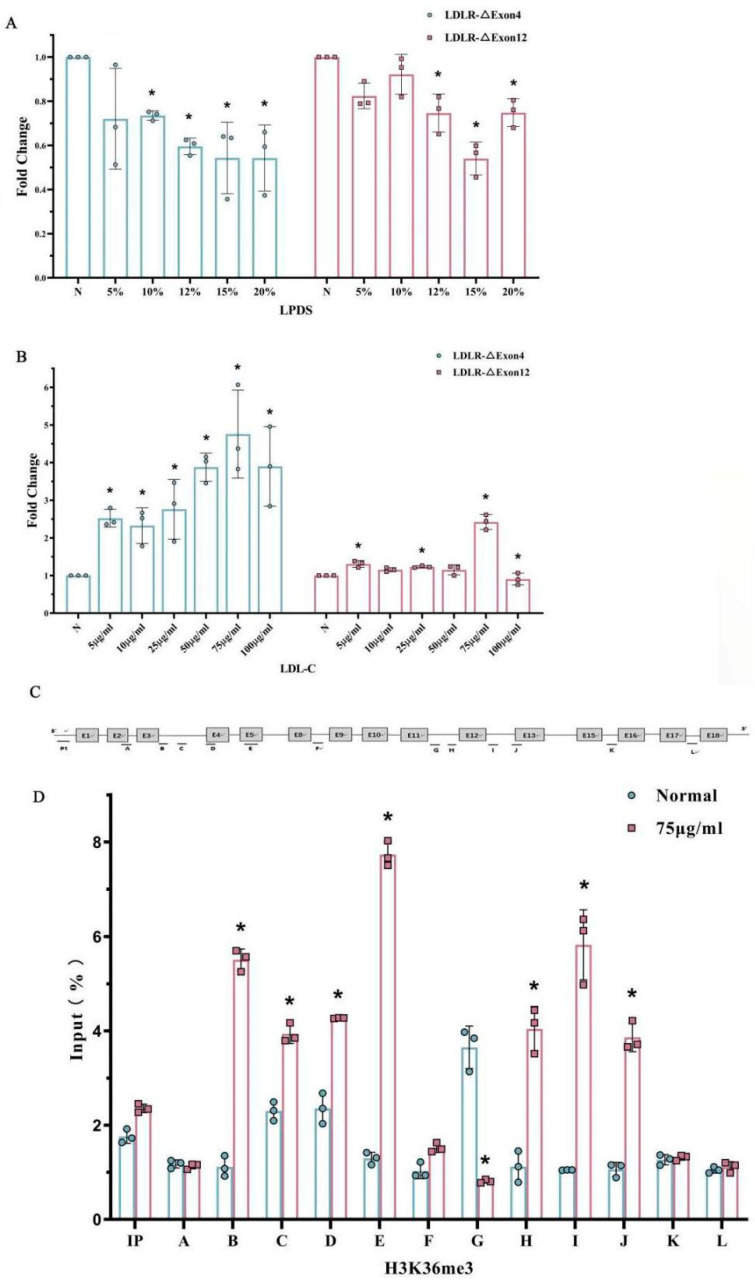
Alternative splicing of LDLR pre-mRNA and histone modification in HepG2 influenced by cholesterol. HepG2 cells were incubated with different concentrations of lipoprotein-deficient serum (LPDS) (Panel (**A**)), or low-density lipoprotein-cholesterol (LDL-C) (Panel (**B**)). Different concentrations of LPDS (Panel (**A**)) and LDL-C (Panel (**B**)) significantly affected the fold changes in both LDLR-∆Exon4 and LDLR-∆Exon12. The nadir of the reduced fold changes in LDLR-∆Exon4 and LDLR-∆Exon12 appeared at 15% of LPDS concentration (Panel (**A**)). The peak of the increased fold changes in LDLR-∆Exon4 and LDLR-∆Exon12 appeared to be associated with 75 μg/mL of LDL concentration (Panel (**B**)). * Indicates a significant difference vs. none (N) treatment (as 1 unit) according to post hoc analysis. Panel (**C**): the schematic structure of primer positions in human LDLR gene. Panel (**D**): The effect of adding 75 μg/mL of LDL-C concentration on relative enrichment of H3-K36me3 in LDLR detected by CHIP assay. Normal: 0 LDL treatment; * *p* < 0.05 vs. 0 LDL-C treatment. Individual data and group mean with SE bars, respectively, are reported.

**Figure 5 ijms-26-04262-f005:**
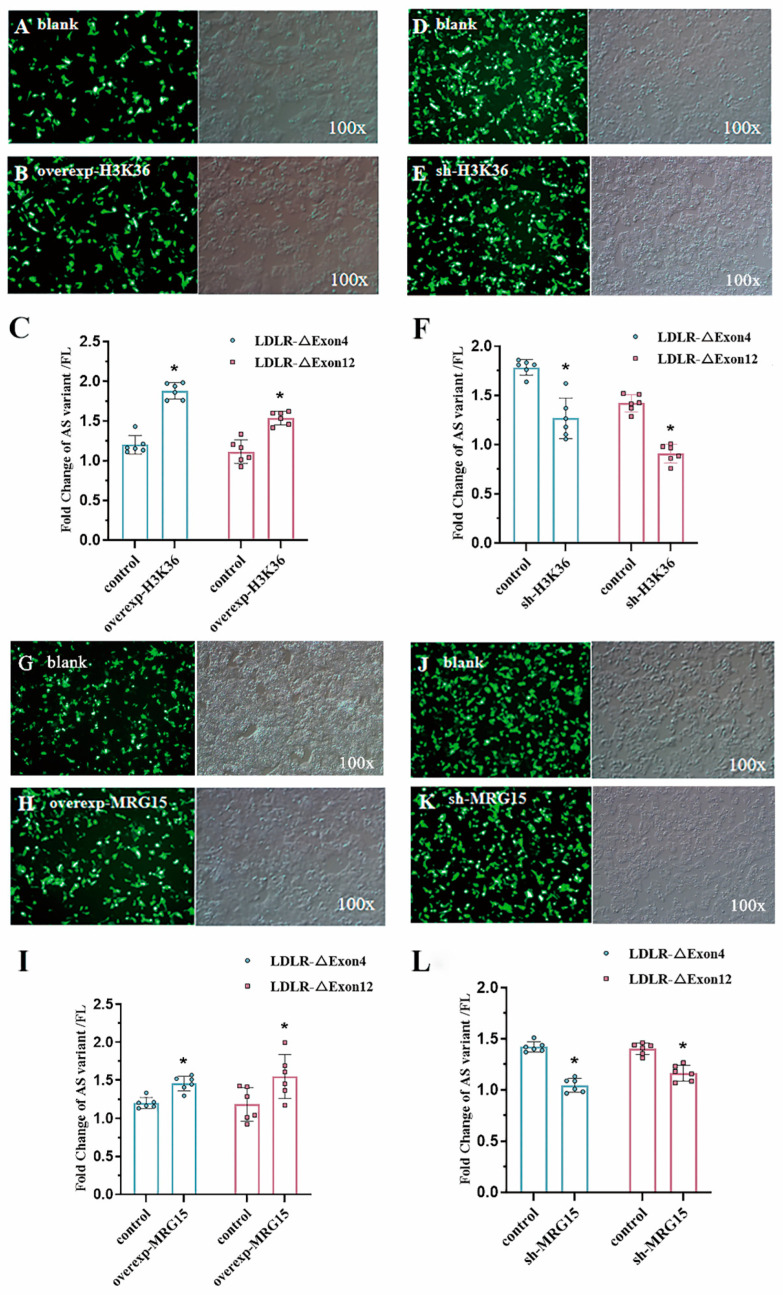
H3K36me3 and MRG15 regulated alternative splicing of LDLR pre-mRNA. Transfection effect of H3K36 methyltransferase overexpression (100×) blank plasmid (Panel (**A**)) vs. overexpression plasmid (Panel (**B**)) on the splicing efficiency of LDLR-∆Exon4 and LDLR-∆Exon12 (Panel (**C**)) in HepG2 cells. Both LDLR-∆Exon4 and LDLR-∆Exon12 were significantly augmented with the overexpressed H3K36 methyltransferase plasmid based on two-tailed Student’s *t*-test. The effect of H3K36 methyltransferase interference (100×) blank plasmid (Panel (**D**)) vs. interference plasmid (Panel (**E**)) on the splicing efficiency of LDLR-∆Exon4 and LDLR-∆Exon12 (Panel (**F**)) in HepG2 cells. Both LDLR-∆Exon4 and LDLR-∆Exon12 were significantly reduced with the interfered plasmid based on two-tailed Student’s *t*-test. Transfection effect of MRG15 methyltransferase overexpression (100×) blank plasmid (Panel (**G**)) vs. overexpression plasmid (Panel (**H**)) on the splicing efficiency of LDLR-∆Exon4 and LDLR-∆Exon12 (Panel (**I**)) in HepG2 cells. Both LDLR-∆Exon4 and LDLR-∆Exon12 were significantly augmented with the overexpressed MRG15 methyltransferase interference (100×) based on two-tailed Student’s *t*-test. Blank plasmid (Panel (**J**)) vs. interference plasmid (Panel (**K**)) on the splicing efficiency of LDLR-∆Exon4 and LDLR-∆Exon12 (Panel (**L**)) in HepG2 cells. Both LDLR-∆Exon4 and LDLR-∆Exon12 were significantly reduced with the interfered MRG15 methyltransferase plasmid based on two-tailed Student’s *t*-test. Individual data and group mean with SE bars, respectively, are presented. * Indicates p < 0.05 vs. control.

**Figure 6 ijms-26-04262-f006:**
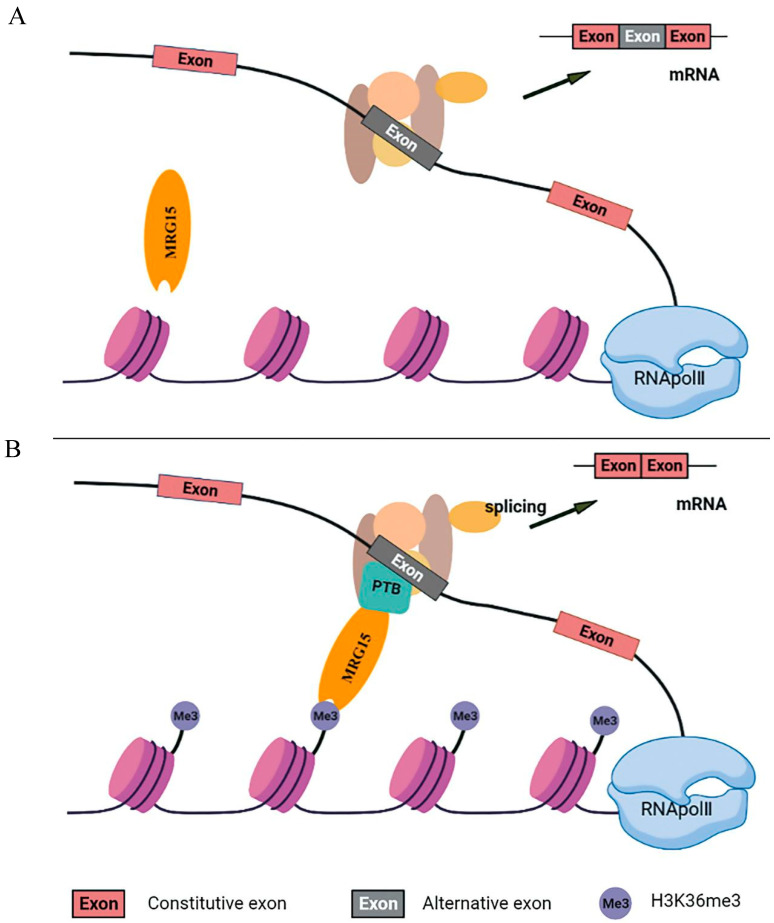
Regulation of LDLR pre-mRNA alternative splicing by histone modification. A proposed adaptor system for reading histone marks by the splicing machinery includes a chromatin-binding protein or MORF-related gene on chromosomes 15 (MRG15), a histone mark signature (H3-K36me3) and a splicing regulator or polypyrimidine tract binding (PTB) protein before (Panel (**A**)) and after it is engaged (Panel (**B**)). MRG15 binds to H3-K36me3 when the histone marks are abundant and recruits PTB which competes with the splicing factor U2AF for binding to the 3′ splice site and converts the H3K36me3 marker into a signal that regulates selective splicing.

**Table 1 ijms-26-04262-t001:** Human physical characteristics and blood lipid baseline data.

	Normal Group (n =10)	Exercise Training Group (n = 10)	High-Cholesterol Group (n = 10)
Men vs. Women	7 vs. 3	7 vs. 3	8 vs. 2
Age (year)	21 ± 2	22 ± 1	22 ± 1
Weight (kg)	73.4 ± 2.8	68.7 ± 3.7 *	90.2 ± 4.3 *^,#^
Height (m)	1.72 ± 0.03	1.75 ± 0.02	1.73 ± 0.03
BMI (kg/m^2^)	24.7 ± 0.5	22.4 ± 0.4	30.5 ± 2.1 *^,#^
MVPA (minutes/week)	406 ± 46	916 ± 86 *	129 ± 42 *^,#^
LDL (mmol/L)	1.67 ± 0.39	0.98 ± 0.39	3.16 ± 0.94 *^,#^
TC (mmol/L)	3.73 ± 0.52	3.15 ± 0.43	6.96 ± 0.58 *^,#^
TG (mmol/L)	1.02 ± 0.25	0.85 ± 0.49	1.88 ± 0.65
HDL (mmol/L)	1.51 ± 0.25	2.52 ± 0.30 *	1.45 ± 0.6 ^#^
AIP	−0.18 ± 0.16	−0.53 ± 0.27 *	0.12 ± 0.30 *^,#^
LDLR-∆Exon4	1.00 ± 0.01	0.72 ± 0.09 *	4.32 ± 0.89 *^,#^
LDLR-∆Exon12	1.00 ± 0.01	0.47 ± 0.33 *	2.96 ± 1.03 *^,#^

BMI: body mass index; MVPA: moderate to vigorous physical activity; LDL: low-density lipoprotein; TC: total cholesterol; HDL: high-density lipoprotein; AIP: atherogenic index of the plasma [calculated from Log(TC/HDL)]; LDLR-∆Exon4: low-density lipoprotein receptor gene mutation at Exon 4; LDLR-∆Exon12: low-density lipoprotein receptor gene mutation at Exon 12. Expression levels of LDLR-∆Exon4 and LDLR-∆Exon12 are normalized to take the levels of the normal group as 1. * a significant difference vs. normal group and ^#^ a significant difference vs. exercise training group according to post hoc analysis.

**Table 2 ijms-26-04262-t002:** The mouse and human LDLR primers used in RT-PCR and CHIP assays.

Gene	Forward Primer	Reverse Primer
Mouse LDLR primers used in RT-PCR		
β-Actin	CATCCGTAAAGACCTCTATGCCAAC	ATGGAGCCACCGATCCACA
LDLR-14(+)	GGTGAACTGGTGTGAGACAACA	AGGAGTACTGGGAGCTGAGAGA
LDLR-14(−)	CCAATCGACTCACGGGTTCA	ACAGTGTCGACTTCTCTAGGC
Human LDLR primers used in RT-PCR		
β-Actin	CTCCATCCTGGCCTCGCTGT	GCTGTCACCTTCACCGTTCC
LDLR-4(+)	TGTCCCCCCAAGACGTGCTCC	CGCAGTTTTCCTCGTCAGATT
LDLR-4(−)	TCAGCTGTGGGGGCCGTGTC	CAGGTGGCCACAGGACAGC
LDLR-12(+)	ATCACCCTAGATCTCCTCAGTG	GCACTGAAAATGGCTTCGTT
LDLR-12(−)	AATGGCATCACCCTAGGACAA	ATCCTCTGGGGACAGTAGGTT
Mouse LDLR primers used in CHIP assays		
GAPDH	AGGTCGGTGTGAACGGATTTG	TGTAGACCATGTAGTTGAGGTCA
P	GGTCCTACCCCCCAAACCAAG	GAAGTATGCGAAGCCCCCTCC
A	TTTGATGATGGATTTGGAAGGTT	TGACAGGTGACAGACACTAGGGA
B	AGCCCCCAAGACGTGCTCCCA	GCCTCGCCGTCACAGACCCAG
C	GGCCCCAACAAGTTCAAGTG	GGCTCATCCGACCAGTCCTG
D	TGACCCCTTTCTTCTGCCTCAGC	TGTCCTCCTCTTTACGCCCTTGG
E	GTGCCAATCGACTCACGGGTT	CGGGGACAAGAGGTTTTCAGC
F	TCTGGCTTACTCTGAAGATGGG	AACTGACAACATTTGATGACGG
G	CAATGGTGGTTGCCAGTACCT	ACTTGTGCCCCTACCTGTGAG
H	ATTCTCATTTTATTTATTCATTT	TGGTATTTCGTTTTCTCCTTCTA
I	ATGCAAGCACTTAGGTGGCG	ACCTCCTCCTAGTCACAACCA
Human LDLR primers used in CHIP assays		
GAPDH	CCATCTTCCAGGAGCGAGAT	CTAAGCAGTTGGTGGTGCAG
P	ATCACCCCACTGCAAACTCC	GATCACGACCTGCTGTGTCC
A	AGTTCCAGTGCCAAGACGG	CAAAGGGGACTCACAGCACG
B	CTGAGTCCTGGGGAGTGGTC	AACCCGAAGAGGTAGCACCA
C	AGTGCTCATAGCAGTGCTGG	TCACCGTGTGAAGTCTCCCA
D	TGACGAGGAAAACTGCGGTA	GGAAATCCACTTCGGCACCTA
E	TGTCGCCCTGACGAATTCCA	GCAGCCAACTTCATCGCTC
F	TCCCGTTGGGAGGTCTTTTCC	AACCAAGAGTGCCTCCCCAT
G	CCTGAAGGTTTCCCTCTTTCTT	TGCAGTGAGCTACGATT
H	GCCTCTCCAGGTGCTTTTCT	GCTGCAGTGAGCTACGATT
I	ATGCAAGCACTTAGGTGGCG	ACCTCCTCCTAGTCACAACCA
J	TGTCATCTTCCTTGCTGCCT	TGACATCGGAACCTGTGAGG
K	CACAACCACCCGACCTGTTC	AAGGGAGTGAGGACGACACC
L	CACAAGGGGTTTAGGGTAGGT	CCAGAGGGAGACGGTGAGTA

## Data Availability

The raw study data supporting the conclusions of this article will be made available by the authors without undue reservation.

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
