# Peer review of "Epigenetic Modifications in Alternative Splicing of LDLR pre-mRNA on Hypercholesterolemia Following Aerobic Exercise Training"

_ijms, 2025, doi:10.3390/ijms26094262_

Round 1
Reviewer 1 Report
Comments and Suggestions for Authors
Epigenetic Modifications in Hypercholesterolemia Following Aerobic Exercise Training
This manuscript explores the epigenetic regulation of LDLR alternative splicing in response to aerobic exercise training, emphasizing the role of H3K36me3 and MRG15 in cholesterol metabolism. The study integrates human, murine, and in vitro models, making it broadly applicable and relevant to cardiovascular health research.
The findings are novel and contribute to our understanding of exercise-induced metabolic reprogramming. However, several missing controls, and unclear statistical analyses must be addressed before acceptance.
Recommendation: Moderate revisions are required.
Weaknesses and Concerns
(A) Language and formatting issues
Line 16-17: The word "young" should be in the same line.
Formatting consistency is required for the following terms:
Line 21: "expression level"
Line 28: "augmented"
Line 32: "alternative"
Lines 119-121, 135-136, 423-426: Ensure consistent formatting.
(B) Experimental design
The human cohort is relatively small (n = 10 per group) and lacks demographic diversity (e.g., age, gender, genetic predispositions).
The mouse study would benefit from a dose-response analysis (e.g., varying durations or intensities of aerobic training).
The HepG2 cell model does not fully recapitulate in vivo hepatic cholesterol metabolism, limiting the generalizability of in vitro findings.
(C) Statistical analysis
ANOVA and Tukey’s post-hoc tests were used for comparisons, but effect sizes were not reported. Including Cohen’s d would strengthen statistical interpretation.
The study lacks correlation analysis between LDLR alternative splicing and lipid profile changes.
(D) Figures and data interpretation issues
Figure 1: LDLR Structural Domains & Alternative Splicing
The figure lacks an explanation of the functional consequences of ΔExon4 and ΔExon12.
Suggestion: Include a brief functional annotation (e.g., "ΔExon4 disrupts LDL binding affinity").
Figure 2: Mouse Experiment Protocol
The figure lacks randomization details (e.g., how mice were assigned to groups).
Suggestion: Indicate whether baseline weight and cholesterol levels were similar before intervention.
Figure 3: Effects of Diet & Exercise on Weight, Lipids, & Liver Histology
No quantitative lipid accumulation analysis in liver tissue.
Missing a loading control (e.g., GAPDH or β-actin) in RT-PCR data.
Figure 4: LDLR Alternative Splicing & Histone Modification in HepG2
No negative control ChIP region (e.g., an unrelated exon or intergenic region).
Lacks direct RNA-binding evidence for the histone effect.
Suggestion: RNA immunoprecipitation (RIP) assay could confirm whether H3K36me3 recruits PTBP1.
Figure 5: Role of H3K36me3 & MRG15 in LDLR Splicing
No rescue experiments (e.g., restoring LDLR function after knockdown).
Suggestion: Use MRG15 overexpression after knockdown to confirm its direct role.
No protein-level validation (Western blot for MRG15).
Suggestion: Include Western blot for MRG15 to validate expression changes.
Figure 6: Proposed Mechanism of H3K36me3-Mediated LDLR Splicing
Lacks direct interaction evidence (e.g., PTBP1 binding to H3K36me3-marked chromatin).
Suggestion: Use a proximity ligation assay.
Lacks validation in primary cells (human hepatocytes).
Suggestion: Test the model in primary liver cells to confirm relevance beyond HepG2.
Conclusion and recommendation
The study presents novel insights into how exercise training modifies LDLR alternative splicing via histone modifications, an understudied area in metabolism research.
However, figures require clarification, particularly regarding controls for epigenetic modifications and splicing efficiency. Additionally, statistical analyses should be expanded, including effect size reporting and correlation analysis between LDLR splicing and lipid levels.
Final recommendation: Moderate revisions required before acceptance.
Comments on the Quality of English Language
Can be improved
Author Response
Reviewer 1
Comments and Suggestions for Authors
Epigenetic Modifications in Hypercholesterolemia Following Aerobic Exercise Training
This manuscript explores the epigenetic regulation of LDLR alternative splicing in response to aerobic exercise training, emphasizing the role of H3K36me3 and MRG15 in cholesterol metabolism. The study integrates human, murine, and in vitro models, making it broadly applicable and relevant to cardiovascular health research.
The findings are novel and contribute to our understanding of exercise-induced metabolic reprogramming. However, several missing controls, and unclear statistical analyses must be addressed before acceptance.
Recommendation: Moderate revisions are required.
Response: We sincerely thank the reviewer for evaluating the manuscript and for the insightful comments and recommendations to help us improve it. Textual revisions made in response to the comments are indicated by blue font in the manuscript text file.
Weaknesses and Concerns
(A) Language and formatting issues
Line 16-17: The word "young" should be in the same line.
Formatting consistency is required for the following terms:
Line 21: "expression level"
Line 28: "augmented"
Line 32: "alternative"
Lines 119-121, 135-136, 423-426: Ensure consistent formatting.
Response: All listed formatting issues are corrected in the revised version of the manuscript. Thank you!
(B) Experimental design
The human cohort is relatively small (n = 10 per group) and lacks demographic diversity (e.g., age, gender, genetic predispositions).
Response: Future studies with large sample size are needed to examine the effects of age with stratifications of gender and ethnicity background. We have addressed this weakness in the revised study limitations and perspectives (see page 20).
The mouse study would benefit from a dose-response analysis (e.g., varying durations or intensities of aerobic training).
Response: We agree that a dose-response of aerobic training - epigenetic impact on cholesterol metabolism in mouse and human would be great.
The HepG2 cell model does not fully recapitulate in vivo hepatic cholesterol metabolism, limiting the generalizability of in vitro findings.
Response: We agree with the reviewer’s point. Primary human hepatocytes are considered a gold standard for in vitro studies investigating the regulatory mechanisms of hepatic glucose, lipid, and cholesterol metabolism. However, access to human primary hepatocytes is often limited, making immortalized hepatic cell models a common alternative. Thus, HepG2 cells have been extensively utilized as an in vitro model for studying cholesterol metabolism, such as to construct a hyperlipidemic hepatocyte model to investigate the role of Anti-b in regulating abnormal TC and TG accumulation in vitro (Bian, Wu et al. 2025) and to validate the mechanism of action of AKG in vitro that effectively attenuates lipid accumulation, ameliorates oxidative stress and regulates mitochondrial function in hepatocytes (Cheng, Zhang et al. 2024). We have included these descriptions in the revised Materials and Methods (see page 25). Thank you!
(C) Statistical analysis
ANOVA and Tukey’s post-hoc tests were used for comparisons, but effect sizes were not reported. Including Cohen’s d would strengthen statistical interpretation.
The study lacks correlation analysis between LDLR alternative splicing and lipid profile changes.
Response: Cohen’s d is designed for between-group effect size for t-test. Since the groups in our human study were more than two groups, we did not provide Cohen’s d for between-group effect size. However, we have calculated and included the correlation coefficient values between the data of LDLR-∆Exon4 and LDLR-∆Exon12 vs LDL-C, TC, and HDL-C in the revised Results (page 8). All correlation coefficients are statistically significant (p < 0.001). Thank you!
(D) Figures and data interpretation issues
Figure 1: LDLR Structural Domains & Alternative Splicing
The figure lacks an explanation of the functional consequences of ΔExon4 and ΔExon12.
Suggestion: Include a brief functional annotation (e.g., "ΔExon4 disrupts LDL binding affinity").
Response: We have added a sentence “Morphological changes in LDLR associated with ΔExon4 and ΔExon12 cause functional impairment of ligand binding domain and EGF precursor domain, respectively” in the revised Figure 1 legend as suggested. Thank you!
Figure 2: Mouse Experiment Protocol
The figure lacks randomization details (e.g., how mice were assigned to groups).
Suggestion: Indicate whether baseline weight and cholesterol levels were similar before intervention.
Response: We have added “Forty male mice were randomly assigned to 4 groups (Figure 2) before the interventions” as the first sentence in The effects of high-cholesterol diet and exercise training on weight and lipid levels in mice. Thank you!
Figure 3: Effects of Diet & Exercise on Weight, Lipids, & Liver Histology
No quantitative lipid accumulation analysis in liver tissue.
Missing a loading control (e.g., GAPDH or β-actin) in RT-PCR data.
Response: The numbers of lipid droplet in liver tissue was examined on HE-stained sections using Image J software. However, only blood plasma lipids and body weight results were reported. For the RT-PCR assays in Figure 3F (ChIP experiments), GAPDH was uniformly used as the loading control, while β-Actin served as the loading control for RT-PCR analysis in Figure 3G. These specifications were annotated in Table 2 of the revised manuscript. Thank you!
Figure 4: LDLR Alternative Splicing & Histone Modification in HepG2
No negative control ChIP region (e.g., an unrelated exon or intergenic region).
Lacks direct RNA-binding evidence for the histone effect.
Suggestion: RNA immunoprecipitation (RIP) assay could confirm whether H3K36me3 recruits PTBP1.
Response:Panels C and D of Figure 4 show unrelated exon regions. We sincerely apologize that we did not perform the RNA immunoprecipitation (RIP) experiment to validate whether H3K36me3 recruits PTBP1 in the present study. Nevertheless, our data seem to provide evidence to support the discussion based on the previous studies reported by Iwamori et al. (Iwamori, Tominaga et al. 2016) who used ChIP-seq experiments to demonstrate strong co-localization of MRG15, PTBP proteins, and H3K36me3 at the exon-intron junctions of the Tnp2 gene and by Luco et al. (Luco, Pan et al. 2010) who confirmed that H3K36me3 recruits MRG15, and PTBP1 to exert its function.
Figure 5: Role of H3K36me3 & MRG15 in LDLR Splicing
No rescue experiments (e.g., restoring LDLR function after knockdown).
Suggestion: Use MRG15 overexpression after knockdown to confirm its direct role.
No protein-level validation (Western blot for MRG15).
Suggestion: Include Western blot for MRG15 to validate expression changes.
Response: We sincerely appreciate these insightful and constructive suggestions. As demonstrated in prior studies, ChIP assays have confirmed the specific recruitment of MRG15 by H3K36me3. Our study primarily focused on elucidating how H3K36me3 regulates LDLR alternative splicing, potentially through MRG15 recruitment. To validate this, we systematically overexpressed or knocked down H3K36me3 and MRG15, demonstrating their regulatory effects on LDLR splicing isoforms. Although these results provide functional evidence, we recognize that further mechanistic studies, including MRG15 protein levels in the nucleus as well as rescue experiments, are needed to fully elucidate this pathway. We plan to conduct more in-depth investigations to further elucidate this regulatory mechanism.
Figure 6: Proposed Mechanism of H3K36me3-Mediated LDLR Splicing
Lacks direct interaction evidence (e.g., PTBP1 binding to H3K36me3-marked chromatin).
Suggestion: Use a proximity ligation assay.
Response: We agree that proximity ligation assay can provide direct evidence for PTBP1 binding to H3K36me3-marked chromatin. Our hypothesis is based on the established H3K36me3-MRG15-PTBP1 cascade regulatory pathway reported in prior studies (e.g., Regulation of alternative splicing by histone modifications) (Luco, Pan et al. 2010, Iwamori, Tominaga et al. 2016). While our current work focuses on the downstream regulatory effects of this pathway on LDLR splicing, we recognize the importance of validating upstream molecular interactions. Future studies will specifically investigate the role of this mechanism in exercise-modulated LDLR alternative splicing, incorporating experimental approaches such as PLA to strengthen the mechanistic evidence.
Lacks validation in primary cells (human hepatocytes).
Suggestion: Test the model in primary liver cells to confirm relevance beyond HepG2.
Response: We fully acknowledge the reviewer's emphasis on the importance of validating the mechanism in primary hepatocytes. Due to the ethical concerns required for obtaining human primary hepatocytes, this study has not yet conducted such validation. However, recent studied demonstrated that atorvastatin (ATS) increased fold enrichment of H3K4me3 at the PCSK9 promoter in the HepG2 model by using HepG2 as a model (Duddu, Katakia et al. 2025) and the κ-keratinase produced KCOs significantly decreased TC, TG and LDL-C levels and increased HDL-C levels in HepG2 cells (Zhu, Mou et al. 2023). These data seem to support the validation of using the classical cell model HepG2 for the relevant studies.
Conclusion and recommendation
The study presents novel insights into how exercise training modifies LDLR alternative splicing via histone modifications, an understudied area in metabolism research.
However, figures require clarification, particularly regarding controls for epigenetic modifications and splicing efficiency. Additionally, statistical analyses should be expanded, including effect size reporting and correlation analysis between LDLR splicing and lipid levels.
Final recommendation: Moderate revisions required before acceptance.
Response: Again, thank you for your insightful suggestions and positive comments.
Literature cited in this response
Bian, Y., H. Wu, W. Jiang, X. Kong, Y. Xiong, L. Zeng, F. Zhang, J. Song, C. Wang, Y. Yang, X. Zhang, Y. Zhang, P. Pang, T. Duo, Z. Wang, T. Pan and B. Yang (2025). "Anti-b diminishes hyperlipidaemia and hepatic steatosis in hamsters and mice by suppressing the mTOR/PPARγ and mTOR/SREBP1 signalling pathways." Br J Pharmacol 182(5): 1254-1272.
Cheng, D., M. Zhang, Y. Zheng, M. Wang, Y. Gao, X. Wang, X. Liu, W. Lv, X. Zeng, K. N. Belosludtsev, J. Su, L. Zhao and J. Liu (2024). "α-Ketoglutarate prevents hyperlipidemia-induced fatty liver mitochondrial dysfunction and oxidative stress by activating the AMPK-pgc-1α/Nrf2 pathway." Redox Biol 74: 103230.
Duddu, S., Y. T. Katakia, R. Chakrabarti, P. Sharma and P. C. Shukla (2025). "New epigenome players in the regulation of PCSK9-H3K4me3 and H3K9ac alterations by statin in hypercholesterolemia." J Lipid Res 66(1): 100699.
Iwamori, N., K. Tominaga, T. Sato, K. Riehle, T. Iwamori, Y. Ohkawa, C. Coarfa, E. Ono and M. M. Matzuk (2016). "MRG15 is required for pre-mRNA splicing and spermatogenesis." Proc Natl Acad Sci U S A 113(37): E5408-5415.
Luco, R. F., Q. Pan, K. Tominaga, B. J. Blencowe, O. M. Pereira-Smith and T. Misteli (2010). "Regulation of alternative splicing by histone modifications." Science 327(5968): 996-1000.
Zhu, C., M. Mou, L. Yang, Z. Jiang, M. Zheng, Z. Li, T. Hong, H. Ni, Q. Li, Y. Yang and Y. Zhu (2023). "Enzymatic hydrolysates of κ-carrageenan by κ-carrageenase-CLEA immobilized on amine-modified ZIF-8 confer hypolipidemic activity in HepG2 cells." Int J Biol Macromol 252: 126401.

Reviewer 2 Report
Comments and Suggestions for Authors
-It is necessary that the title of Table 1 clarifies that baseline data are being presented. I suggest that this table be completed with post-training data.
-I suggest describing the diet of the groups because in addition to exercise, the type of diet can influence the concentration of lipids in the blood.
- Indicate whether overweight volunteers had familial dyslipidemia
-I suggest presenting the calculation of the atherogenic index and atherogenic index of the plasma.
-The time for dyslipidemia to be observed is approximately 30 days; so I ask if the authors considered doing the follow-up at 30 days.
-For the discussion, I suggest addressing the loss of the protective effect of exercising for two months.
Author Response
We sincerely thank the reviewer for evaluating the manuscript and for the positive and insightful comments and recommendations to help us improve it. Textual revisions made in response to the comments are indicated by blue font in the manuscript text file.
Reviewer 2
Comments and Suggestions for Authors
-It is necessary that the title of Table 1 clarifies that baseline data are being presented. I suggest that this table be completed with post-training data.
Response: We fully concur with the reviewer's suggestion that Table 1 should be clearly labeled as "baseline data". As the current analysis focuses solely on cross-sectional comparisons among young adults with normal cholesterol levels, hyperlipidemic non-exercisers, and hyperlipidemic exercisers, there are no post-training data.
-I suggest describing the diet of the groups because in addition to exercise, the type of diet can influence the concentration of lipids in the blood.
Response: We fully acknowledge that dietary choice of young subjects could significantly effect on blood lipid concentrations, in addition to lifestyle. This seems to be supported by the data on mouse study of the present study. However, during the initial design phase of this study, our primary focus was on investigating the impact of exercise interventions on cholesterol levels and alternative splicing of LDLR in human study. We were unable to systematically collect participants' dietary information, especially those subjects who visited blood bank. We recognize this as a limitation of our study.
- Indicate whether overweight volunteers had familial dyslipidemia
Response: The absence of systematic medical archives at the blood bank center, restricting the accuracy of retrospective family history tracing. In follow-up studies, we will implement a structured family history questionnaire to address this issue.
-I suggest presenting the calculation of the atherogenic index and atherogenic index of the plasma.
Response: The atherogenic index of plasma (AIP) has now been calculated using the formula AIP = log(TG/HDL-C), and included in the revised Table 1. Thank you!
-The time for dyslipidemia to be observed is approximately 30 days; so I ask if the authors considered doing the follow-up at 30 days.
Response: Certainly, it is a great idea to have the follow-up observation in future study. We will definitely consider this.
-For the discussion, I suggest addressing the loss of the protective effect of exercising for two months.
Response: Initially, our study did not include aerobic exercise interventions for young patients with hypercholesterolemia, so the potential loss of protective effects after two months of exercise cessation was not addressed in the experimental design. Your suggestion is highly valuable. In subsequent experiments, we will investigate both the interventional effects of aerobic exercise on hypercholesterolemia and the decay of exercise-induced protective effects after derating.
Again, thank you for your positive and constructive comments and suggestions which have helped us strengthen the manuscript.

Round 2
Reviewer 1 Report
Comments and Suggestions for Authors
The modifications made are satisfactory.
Author Response
Reply to Academic Editor Notes
In addition to the comments submitted by the reviewers, additional corrections should be made:
The English needs improvement (e.g., page 23: “how many days a seek” ? or not “climb stairs” but climbing )
Response: These typos are corrected, and the English of whole manuscript has been double checked.
Page 4 - Sentence: „The LDLR pathway is the most vital link in cholesterol metabolism by taking up cholesterol-rich lipoproteins in the circulation, preventing their accumulation in the bloodstream, and facilitating the ingested cholesterol entering the cells to promote cell proliferation and the synthesis of steroid hormones.”
- Cholesterol-rich lipoproteins include LDL and HDL. The LDL receptor is responsible for LDL uptake, not HDL uptake.
Response: It is changed to “The LDLR pathway is the most vital link in cholesterol metabolism by taking up cholesterol-rich low-density lipoprotein (LDL-C) in the circulation” (page 4).
- LDL transports cholesterol deriving from both diet and biosynthesis. In addition, dietary cholesterol is not ingested but absorbed. The use of the phrase “ingested cholesterol” is not proper.
Response: “ingested” is replaced with “absorbed” (page 4). Thank you!
Page 5 – In this manuscript, LDL-C means LDL-cholesterol. Therefore, it is not proper to write” where LDLR-ΔExon4 affects the affinity for binding to the ligand LDL-C”, as LDL-cholesterol is not the ligand of the LDL receptor. The LDL protein - Apo B is the ligand.
Response: It is changed from “where LDLR-∆Exon4 affects the affinity for binding to the ligand LDL-C” to “where LDLR-∆Exon4 affects the affinity via Apolipoprotein B for binding to the ligand LDL-C” (page 5).
Page 6- “Abnormality in the LDLR gene” - It is necessary to add what kind of abnormality
“ … cholesterol-rich lipoproteins …” – improper wording as indicated before
Response: “… cholesterol-rich lipoproteins…” is replaced with “… LDL-C …”.
“Aerobic exercise seems to be emerging as an effective non-pharmaceutical preventive and therapeutic intervention for hypercholesterolemia.” There is no clinical data to support this statement, as there is no data to suggest that aerobic exercise can significantly reduce cholesterol levels in familiar hypercholesterolaemia. Aerobic exercise may improve lipoprotein metabolism and lead to a reduction in serum cholesterol levels and may be effective in correcting disturbances in lipoprotein metabolism. However, it is difficult to accept that aerobic exercise can significantly reduce serum cholesterol levels in familial hypercholesterolaemia.
“improve hypercholesterolemia in gene expression” – What does it mean?
Response: “Aerobic exercise seems to be emerging as an effective non-pharmaceutical preventive and therapeutic intervention for hypercholesterolemia” is changed to “Aerobic exercise seems to be emerging as an effective non-pharmaceutical preventive and therapeutic intervention for improving metabolism of hypercholesterolemia.” In addition, the sentence “The objective of the present study was to investigate aerobic exercise training induced epigenetic modification to improve hypercholesterolemia in gene expression…” is changed to “The objective of the present study was to investigate aerobic exercise training induced epigenetic modification to improve metabolism of hypercholesterolemia in gene expression of LDLR” (page 6). Thank you!
In general, the introduction should be shortened. There is no need to present information on hypocholesterolemic drugs as they are not the subject of the publication, and this information does not add anything new.
Response: We have deleted the descriptions of the information on drugs in the revised Introduction (page 5) as suggested.
Page 3 – “2.1. The effect of habitual physical activity on the alternative splicing of LDLR pre-mRNA in human”
- This subtitle needs to be changed as the presented data indicate differences between studied groups, but different factors can be responsible for observed differences.
- There is no need to describe the differences in anthropometric and biochemical parameters between the groups in such detail. A brief description should be provided indicating that these groups differ significantly from each other. It should be noted that due to the small number of groups, these results should be considered preliminary.
Response: This subtitle is changed to “The effect of habitual physical activity on lipid levels and LDLR in humans”. Furthermore, descriptions on BMI and body fat have been eliminated.
Table 1
- The number of men and women should be presented.
- - It is not appropriate to present mean waist-to-hip ratio and body fat % data for the whole group of women as men because of anatomical differences. Such aggregate data are not meaningful.
Response: Table 1 is revised – the number of men vs women included and waist-to-hip ration and body fat % removed. There is no statistical difference (P = 0.619) in the gender proportions among the groups according to Mantel-Haenszel Chi-Square test.
The discussion should be improved, and it should be very clear which data are from human studies and which are from animal studies. The current description is chaotic and may mislead the reader. Many elements are repeated in different parts of the discussion. In addition, the discussion often takes the form of a review publication, reporting various well-known data that do not add new information. The discussion should not repeat the information given in the introduction or describe the results and refer directly to them. The discussion should focus on highlighting relationships and differences with the results of other authors and suggesting directions for future research. It should be emphasised that the human studies presented involve very small groups and should be considered preliminary studies.
Response: We have emphasized that our human study data are based on preliminary cross-sectional study (see page 5). Thank you so much for your expert notes which help improve the manuscript!
